DATA RELEASE

# Chromosome-scale assembly of the highly heterozygous genome of red clover (*Trifolium pratense* L.), an allogamous forage crop species

Derek M. Bickhart[1],[*], Lisa M. Koch[1], Timothy P. L. Smith[2], Heathcliffe Riday[1] and Michael L. Sullivan[1],[*]

1  US Dairy Forage Research Center, United States Department of Agriculture Agricultural Research Service (USDA-ARS), Madison, WI, USA
2  US Meat Animal Research Center, United States Department of Agriculture Agricultural Research Service (USDA-ARS), Clay Center, NE, USA

## ABSTRACT

Relative to other crops, red clover (*Trifolium pratense* L.) has various favorable traits making it an ideal forage crop. Conventional breeding has improved varieties, but modern genomic methods could accelerate progress and facilitate gene discovery. Existing short-read-based genome assemblies of the ~420 megabase pair (Mbp) genome are fragmented into >135,000 contigs, with numerous order and orientation errors within scaffolds, probably associated with the plant's biology, which displays gametophytic self-incompatibility resulting in inherent high heterozygosity. Here, we present a high-quality long-read-based assembly of red clover with a more than 500-fold reduction in contigs, improved per-base quality, and increased contig N50 by three orders of magnitude. The 413.5 Mbp assembly is nearly 20% longer than the 350 Mbp short-read assembly, closer to the predicted genome size. We also present quality measures and full-length isoform RNA transcript sequences for assessing accuracy and future genome annotation. The assembly accurately represents the seven main linkage groups in an allogamous (outcrossing), highly heterozygous plant genome.

**Subjects**  Genetics and Genomics, Bioinformatics, Plant Genetics

**Submitted:**  27 October 2021

*  Corresponding authors. E-mail: derek.bickhart@usda.gov; michael.sullivan@usda.gov

Preprint submitted at https://doi.org/10.1101/2022.01.06.475143

# DATA DESCRIPTION

## Background

The species *Trifolium pratense* L. (red clover, NCBI:txid57577) is an important legume forage crop grown on approximately 4 million hectares worldwide [1]. Red clover is a versatile crop grown as animal feed and/or as a green manure in pure and mixed stands for hay, haylage, silage, and grazing. Red clover is known for its ease of establishment and shade tolerance, and its ability to grow in poorly drained and low pH soils. The reduced need for exogenous nitrogen application owing to its nitrogen-fixing ability and the relatively high protein content of this plant compared with other forage crops provide potential for reducing the environmental footprint of livestock production. Compared to alfalfa, another common legume forage crop, red clover varieties have higher forage yields, are a better source of magnesium to avoid grass tetany in grazing cattle, and may have improved post-harvest protein preservation [2] and bypass protein content in ruminant production

systems [3]. The improved protein storage and utilization of this forage appears to be associated with the post-harvest oxidation of *o*-diphenolic compounds by an endogenous polyphenol oxidase [4], although condensed tannins could also play a role [5]. Red clover tissues accumulate polyphenol oxidizable phenolics (mainly caffeic acid derivatives), condensed tannins, and various specialized metabolites, including flavonoid compounds [6, 7]. Such compounds can influence animal and rumen physiology both negatively [8] and positively [9]. Specialized metabolites from red clover also have potential medicinal or nutraceutical value (see, for example, [10]). Improved red clover varieties have been developed, especially with respect to persistence, disease resistance, and yield, but further improvements could be made in these and other traits affecting quality and nutritional value [1]. Genetic progress and greater understanding of the physiology and biochemistry of agronomic and quality traits could be accelerated using genomic tools based on the production of a high-quality reference genome for the species. Such a genome would also facilitate gene discovery efforts.

## Context

Red clover is a hermaphroditic allogamous (outcrossing) diploid ($2n = 2x = 14$) with a homomorphic gametophytic self-incompatibility (GSI) system [11] whereby a pistil expressed S-RNase mediates the degradation of pollen tubes from "self" pollen [12]. The GSI locus has been mapped to linkage group one in red clover. The GSI system in red clover appears to be especially effective [13], making red clover an obligate out-crossing species with a high degree of heterozygosity. This high degree of heterozygosity has made genome assembly with short-read sequencing data difficult. Two previous short-read genome assemblies [14, 15] have been reported with limited contiguity (>135,000 contigs), completeness, and accuracy. We report a long-read based assembly consisting of 258 contigs, which provides a much-improved reference genome to enhance genome-enabled red clover improvement.

## METHODS

### Sample information

The individual used in this study is HEN17-A07, a red clover plant selected from the US Dairy Forage Research Center (Madison, WI, USA) breeding program, representing elite North American red clover germplasm (Figure 1). This individual derived from 30 years of selection and breeding for red clover grazing tolerance, persistence, biomass yield, and *Fusarium oxysporum* Schlect resistance [16, 17]. Source varieties and germplasm for HEN17-A07 include the red clover varieties "Dominion" [18] and "Redlangraze" (ABI Alfalfa Inc., now part of Land O'Lakes, Inc. Arden, MN, USA); and experimental populations C452, C11, and C827 out of the US Dairy Forage Research Center red clover breeding program. Plant material used for all nucleic acid isolations was clonally propagated from the original selected plant and maintained in a growth chamber at 22 °C with 18-h days and light intensities of approximately 400 μmol m$^{-2}$ s$^{-1}$.

### DNA and RNA extraction and sequencing

Approximately 0.8 g of frozen unexpanded leaf tissue from the red clover individual Hen17-A07 (hereafter referred to as "red clover") was ground in a mortar and pestle under liquid nitrogen. High-molecular-weight DNA was extracted using the NucleoBond HMW

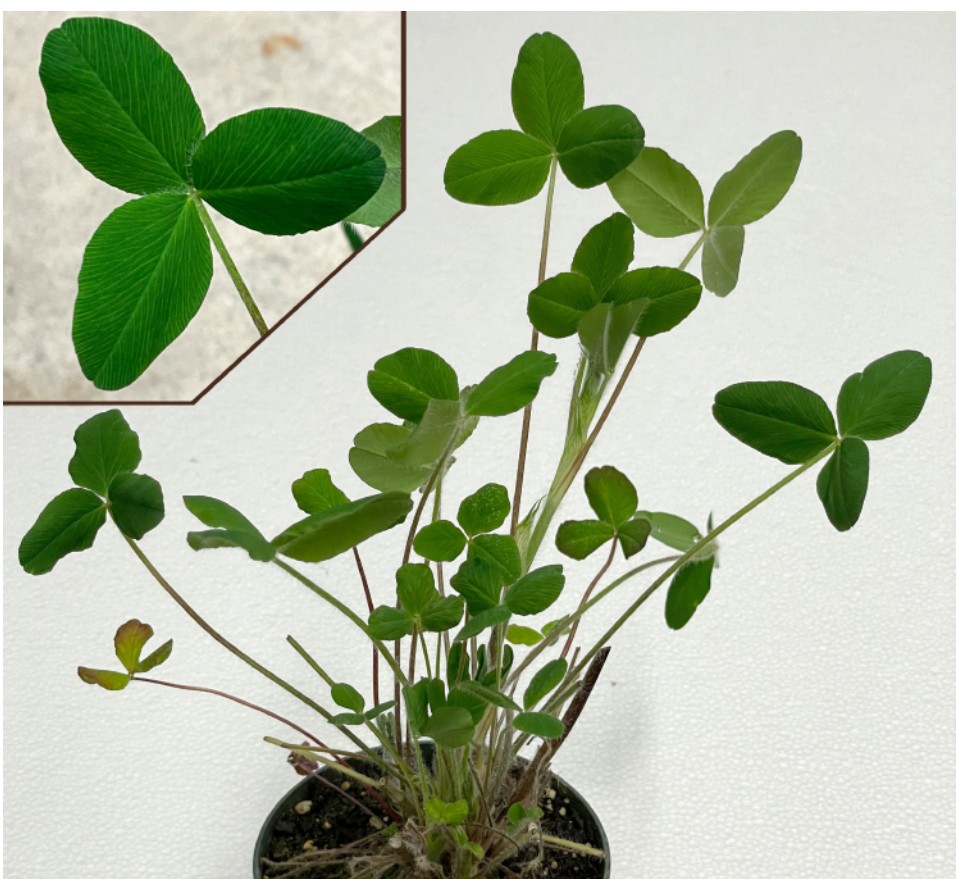

**Figure 1.** *Trifolium pratense* L. HEN17-A07. This image shows a HEN17-A plant that was clonally propagated from the sequenced individual. Leaf detail is shown (inset).

DNA extraction kit as directed by the manufacturer (Macherey Nagel, Allentown, PA, USA). The DNA pellet was resuspended in 150 µL of 5-mM Tris-Cl pH 8.5 (kit buffer HE) by standing at 4 °C overnight, with integrity estimated by fluorescence measurement (Qubit, Qiagen, Germantown, MD, USA), optical absorption spectra (DS-11, DeNovix), and size profile (Fragment Analyzer, Thermo Fisher, Waltham, MA, USA).

The Ligation Sequencing Kit (SQ*K*-LSK109) was used to prepare libraries for Nanopore sequencing from the extracted DNA as directed by the manufacturer (Oxford Nanopore Technologies, Oxford, UK). The libraries were sequenced in 14 R9.4 MinION flowcells on a GridION ×5 instrument. The Guppy version 3.3 basecaller was used to call sequence bases producing 60 gigabase pairs (Gbp) of nanopore sequence in 4.5 million pass_filter reads, having average read length of 13.6 kilobase pairs (Kbp).

The DNA for HiFi sequencing was sheared (Hydroshear, Diagenode, Denville, NJ, USA) using a speed code setting of 13 to achieve a size distribution with peak at approximately 23 Kbp. Smaller fragments were removed by size selection for >12 Kbp fragments (BluePippin, Sage Science, Beverly, MA, USA). Size-selected DNA was used to prepare a SMRTbell library using the SMRTbell Express Template Prep Kit 2.0 as recommended by the manufacturer (Pacific Biosciences, Menlo Park, CA, USA). The library was sequenced in two SMRT Cell 8M cells on a Sequel II instrument using Sequel Sequencing Kit 3.0, producing 23.2 Gbp of HiFi sequence in 1.22 million CCS reads having average length 18.9 kb.

Approximately 200 μg of DNA was fragmented to approximately 550 bp on a Covaris M220 focused-ultrasonicator (Covaris, Woburn, MA, USA) by the University of Wisconsin–Madison Biotechnology Center (Madison, WI, USA) for short-read sequencing as specified in the TruSeq DNA PCR-Free Reference Guide [19]. A library was prepared using a TruSeq DNA PCR-Free library preparation kit, according to manufacturer guidance, and was sequenced on a NextSeq 500 instrument (Illumina) with a NextSeq High Output v2 300 cycle kit, generating 198 million 2 × 150 paired end reads. This resulted in 30.0 Gbp of short-read data, which were used for error-correction and assembly validation.

The Omni-C library was prepared from unexpanded leaf tissue collected from plants grown in the dark for 3 days, and ground in liquid nitrogen with a mortar and pestle. The pulverized material was processed into a proximity ligation library using the Omni-C Proximity Ligation Assay Protocol of the Omni-C Kit, as directed by the manufacturer (Dovetail Genomics, Scotts Valley, CA, USA). The library was sequenced on a NextSeq 500 instrument (Illumina) with 2 × 150 paired-end reads, generating 60 million paired-end Hi-C reads.

RNA was prepared for Iso-seq using the Sigma Spectrum Plant Total RNA Kit including On-Column DNAse I Digestion (both Sigma–Aldrich, St. Louis, MO, USA). One Hen17-A07 plant was sectioned into three parts (roots, leaves/crown, stem/flower), which were ground separately in liquid nitrogen in a mortar and pestle. RNA was prepared from 100 mg of each of the three tissues and pooled in equal proportions to avoid overrepresentation of one portion of the plant in the Iso-seq reads. Pooled RNA was processed into an Iso-seq library using the "Iso-Seq Express Template Preparation for Sequel and Sequel II Systems" protocol from the manufacturer [20] using the "standard" workflow of the protocol, which includes a selection for polyadenylated transcripts. The library was sequenced in four SMRT cells on a Sequel II instrument, producing a total of 49 million subreads, with an average length of 2.9 Kbp.

## Genome assembly and scaffolding

HiFi reads (23.2 Gbp total; approximately 55–60× predicted coverage) were assembled using the PacBio Improved Phased Assembler (IPA) HiFi assembler version 1.3.0 [21] using default settings. This resulted in a primary haplotype assembly of 419.1 Mbp in 283 contigs, with a contig N50 of 4.3 Mbp, and an alternate haplotype assembly of 353.6 Mbp in 1555 contigs. The relatively large size of the alternate haplotype assembly likely reflects the obligate heterozygosity of red clover, since high heterozygosity supports more complete separation of parental haplotypes during HiFi-based assembly. The primary haplotype assembly was retained for use in downstream polishing and assembly quality assessment. Residual haplotype sequence was removed from the assembled contigs using purge_dups v1.2.5 (RRID:SCR_021173) [22]. Depth of coverage cutoff values for the purge_dups workflow were estimated from minimap2 (RRID:SCR_018550) [23] alignments of HiFi reads to the contigs. A total of 5.6 Mbp (1.4% of the original bases) in 34 contigs were identified as remnant haplotypes in the primary contig assembly and removed. Of the 34 contigs, 25 were entirely composed of remnant haplotype sequence and were completely removed from the purged assembly. The final set of purged contigs (hereafter referred to as "HiFi Contigs") had an identical contig N50 (4.3 Mbp) to the first primary IPA assembly because of the small size of the contigs that were removed, but had 258 contigs and a reduction in size of 5.6 Mbp.



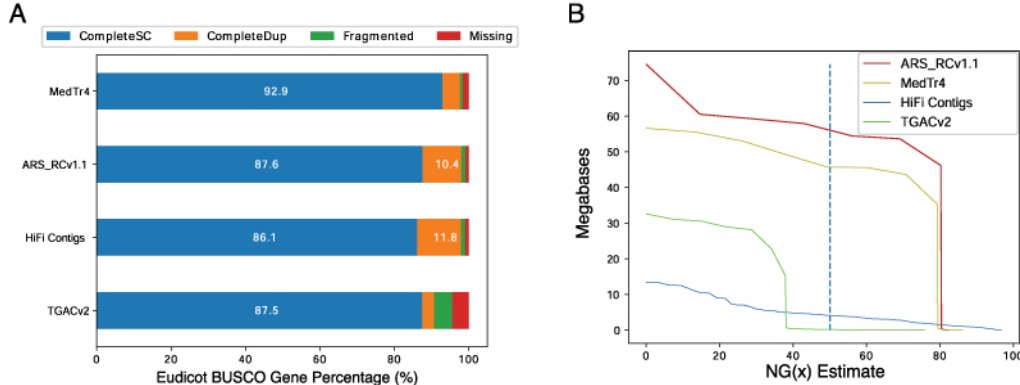

**Figure 2.** Comparative assembly statistics. (A) The total percentages of Eudicot lineage single-copy orthologous genes identified by the BUSCO tool are represented by stacked histograms for each assembly. Values larger than 10% are displayed on the histograms for convenience. (B) NG values against an estimated genome size of 420 Mbp are shown as solid lines on the plot. The NG50 value is distinguished by a vertical dashed bar for each assembly.

**Table 1.** Assembly size statistics.

| Statistic | TGACv2 | HiFi contigs | Omni-C scaffolds | MedTr4 |
|---|---|---|---|---|
| Assembly length (Mbp) | 346.0 | 413.5 | 413.5 | 412.8 |
| Contig/scaffold count | 39,051 | 258 | 143 | 2,186 |
| Scaffold N50 (Mbp) | 22.7 | 4.4 | 15.6 | 49.2 |
| Largest contig/scaffold (Mbp) | 32.6 | 13.4 | 34.2 | 56.6 |

Scaffolds were created from the HiFi Contigs using the SALSA v2 scaffolding workflow [24]. Omni-C reads were aligned to the purged contig assembly using BWA MEM [25] with the "-SP5" flag to disable paired-end read recovery. Resulting BAM files were converted to a bed file format using the Bedtools2 (RRID:SCR_006646) [26] tool "bamToBed". SALSA was subsequently run without misassembly detection to avoid unnecessary contig breaks and the "DNASE" setting owing to the use of OmniC reads for scaffolding. This placed the 258 contigs into 143 scaffolds with a scaffold N50 of 15.6 Mbp (Table 1). This intermediary dataset is referred to as "Omni-C scaffolds" for convenience. The contiguity, as summarized by the contig and scaffold N50 values, compared favorably with legume assemblies that had the benefit of extensive polishing, such as the *Medicago truncatula* reference, MedTr 4.0 [27] (Figure 2).

## Scaffold placement using linkage data

Previously published expressed sequence tag (EST) [28], bacterial artificial chromosome (BAC) end [14], and Oxford Nanopore read overlaps were used to generate super-scaffolds representing the best approximation of red clover linkage group chromosomes. EST and BAC reads were converted to fasta format and aligned against Hi-C scaffolds using BWA MEM. A custom script [29] was used to order and orient EST and BAC information into a tabular, bipartite graph format for comparison. Oxford Nanopore reads were aligned to the Omni-C scaffolds with minimap2 [23] and overhanging reads were identified using custom Perl scripts [30]. Overlapping reads from two different contigs were combined into bipartite graphs as evidence of connection.

BAC, EST, and Oxford Nanopore datasets were analyzed using the Python NetworkX (version 2.5) [31] module to determine concordance among all three for final scaffold

formation. The Oxford Nanopore read overlaps showed substantial overlap with the underlying EST dataset, but the BAC end sequence showed no substantial overlap with other datasets. The final linkage group super-scaffolds were generated by assigning Omni-C scaffolds to linkage groups and ordering them according to their placement in the EST alignment dataset. Scaffolds that did not have EST mappings but were identified via Nanopore overlaps (four scaffolds in total) were incorporated into the final super-scaffolds on the side of the scaffold indicated by overlapping read data. The final set of super-scaffolds were generated using the "agp2fasta" utility of the "CombineFasta" Java tool (version 0.0.17) [32]. The final set of super-scaffolds is referred to as "ARS_RCv1.1" in the text.

## Iso-seq transcript identification

Iso-seq sequence data was processed for isoform identification using the Iso-Seq Analysis pipeline in smrtlink v9.0.0.92188, including the option to map putative isoforms to the assembly scaffolds imported as a reference genome. A total of 9.2 million HiFi reads were generated from the 49 million sub-reads, of which 8,899,606 (97%) were classified as full-length non-concatemer reads (FLNC) with a mean length of 3.2 Kbp. These FLNC reads collapsed to 437,586 predicted unique polished high-quality isoforms, of which 308,804 (70%) mapped to 24,955 unique gene loci in the assembly, consistent with approximately 12 isoforms per unique loci. These gene loci are provided as BED coordinate files for future annotation efforts.

## DATA VALIDATION AND QUALITY CONTROL

## Assembly error-rate assessment

Genome quality was tested using a composite of $k$-mer and read mapping quality statistics as implemented in the Themis-ASM workflow [33]. All references to short-read whole genome sequence (WGS) data refer to the short-reads generated from the HEN17-A07 individual sequenced and assembled in this study unless otherwise mentioned. The completeness and quality of the assembly was first assessed using Merqury [34] $k$-mer analysis and FreeBayes (RRID:SCR_010761) [35] variant analysis. Merqury estimated the overall quality of the assembly at a Phred-based [36] quality value (QV) score of 49, which corresponds to an error every 129,000 bases (Table 2). Comparison of $k$-mer profiles between the HiFi contigs and the previously published TGACv2 red clover assembly [14] (accession GCA_900292005.1) using the -Py-UpSet Python module [37] (Figure 3) indicated that only 55.2% of all $k$-mers were shared between the two assemblies. This surprisingly low shared content could be the result of real differences in the genomes of the different varieties of this highly heterozygous species (the earlier assembly used an individual from the Milvus variety versus the Hen17-A07 individual used here), or the higher degree of completeness of the current assembly (the previous assembly comprised 135,023 contigs and was 68 Mbp smaller), or assembly and haplotype switching errors in the short read assembly, or a combination of these factors. The Themis-ASM analysis of TGACv2 estimated an error every 142 bases, indicating that the ARS_RCv1.1 assembly has a three orders of magnitude improvement in $k$-mer-based QV estimates. Indeed, the count of erroneous, singleton $k$-mers identified in the TGACv2 assembly was over 40 million, compared to less than 10,000 in the ARS_RCv1.1 assembly (Figure 4). This represents a substantial improvement in assembly accuracy enabled by improved sequencing technologies.

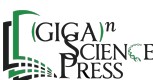

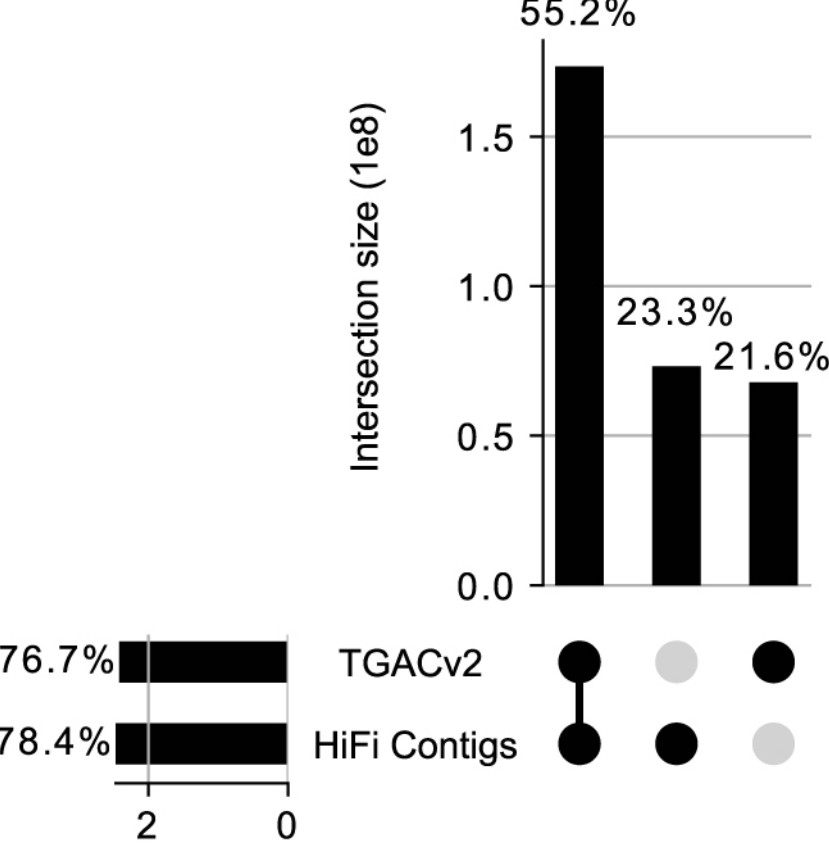

**Figure 3.** Comparison of unique *k*-mer counts in the TGACv2 assembly and our HiFi contigs. Unique *k*-mers were counted using meryl and compared between both assemblies using exact match comparisons. The top histogram shows the proportion of all unique *k*-mers shared by each set, with set membership shown in the bottom right dot plot. The leftmost histogram shows the total count of unique *k*-mers distinct to each assembly, with percentages indicating the amount of *k*-mers from the combined total dataset.

**Table 2.** Assembly quality statistics.

| Category | TGACv2 | ARS_RCv1.1 | MedTr4 | Description |
|---|---|---|---|---|
| Merqury QV | 21.5304 | 48.9101 | 9.74458 | *k*-mer-based quality |
| Merqury ErrorRate | 0.007 | $1.29 \times 10^{-5}$ | 0.106 | *k*-mer-based error rate |
| Merqury completeness (%) | 61.7428 | 77.7322 | 3.86382 | Percentage of complete assembly based on *k*-mers |
| Freebayes QV | 20.03 | 41.71 | 12.22 | SNP and INDEL quality value |
| Unmapped reads (%) | 3.65 | 2.37 | 60.92 | Percentage of short-reads unmapped |
| COMPLETE single copy (%) | 87.5 | 87.6 | 92.9 | Percent of complete, single-copy BUSCOs |
| COMPLETE duplicated (%) | 3.2 | 10.4 | 4.8 | Percent of complete, duplicated BUSCOs |
| FRAGMENTED (%) | 4.9 | 1.1 | 0.7 | Percent of fragmented BUSCOs |
| MISSING (%) | 4.4 | 0.9 | 1.6 | Percent of missing BUSCOs |

FreeBayes QV values were similar to those generated via Merqury analysis, but with a six-point decrease in relative QV between the two assemblies. This QV estimate was originally developed to compare the qualities of uniquely mappable regions of assemblies [38], so it is more robust when comparing datasets derived from different breeds or varieties to separate assemblies. The appreciable difference in FreeBayes QV

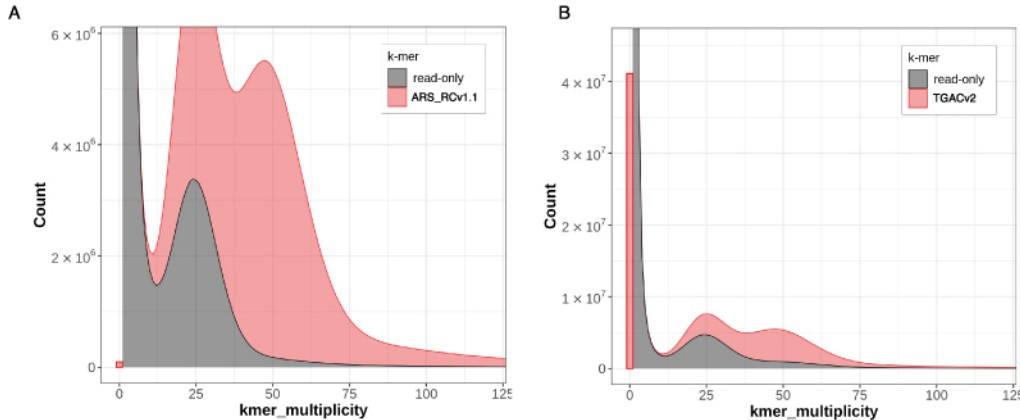

**Figure 4.** Merqury stacked histogram charts of *k*-mer multiplicity between the ARS_RCv1.1 (A) assembly and the TGACv2 (B) reference. In each case, the *k*-mers derived from the assembly are colored light red, and the *k*-mers unique to the short-read whole genome sequence (WGS) data (from the HEN17-07A individual of *Trifolium pretense*) are dark gray. The farthest left red bar indicates the total number of singleton *k*-mers for each assembly, which are considered indicators of misassemblies or errors. The bimodal distribution of each plot indicates the heterozygous (left-most) and homozygous (right-most) *k*-mer values. The prevalence of any area under the "read-only" plot indicates that the assembly does not contain *k*-mers present in the short-read WGS data.

between the two assemblies still points towards a higher error rate in the TGACv2 reference, and suggests that the ARS_RCv1.1 assembly is more suitable as a reference for short-read WGS alignment in the red clover species. The MedTr4 assembly represents a high-quality reference for most legume species, and has been used in several whole genome comparisons to indicate assembly quality [39, 40]. This includes the original release of the TGACv2 reference, where synteny was identified between MedTr4 and the TGACv2 assembly [14]. However, the Merqury-estimated error rate of one out of every ten bases when mapping red clover WGS reads suggests that MedTr4 is unsuitable as a reference for red clover WGS alignment. This conclusion is supported by the observation that over 60% of the HEN17-A07 individual WGS reads were unmapped when aligned to the MedTr4 reference. This suggests that more distantly related legume species require a high-quality reference genome assembly for satisfactory alignment quality metrics. The approach described here provides a method to develop these reference assemblies for highly heterozygous allogamous species, such as red clover, without the requirement for extensive *post-hoc* polishing.

## Structural variant assessment and comparative alignments

The structural accuracy of the super-scaffolds was assessed using a variety of comparative metrics native to the Themis-ASM workflow [33]. The short-read WGS data alignments were used as a basis for FRC_align [41] quality metrics, which identified a relatively low number of regions with predicted interscaffold alignments in ARS_RCv1.1 (Table 3). This was matched by a relatively low count of complex structural variants (SV) in ARS_RCv1.1 compared with TGACv2, as identified by Lumpy [42] analysis, suggesting that small-scale misassemblies that are detectable using short-read alignments were minimized in the ARS_RCv1.1 assembly.

Comparisons of the large-scale synteny of our assembly to the TGACv2 reference revealed a substantial number of discrepancies. Alignment of the scaffolds from the



**Table 3.** Structural variant analysis

| Category | TGACv2 | ARS_RCv1.1 | Description |
|---|---|---|---|
| HIGH_SPAN_PE | 65,254 | 2,052 | FRC_align identified regions with high numbers of inter-contig paired-end read mappings |
| Lumpy deletions | 20,727 | 20,945 | Number of identified structural variant deletions |
| Lumpy duplications | 6,554 | 3,823 | Number of identified structural variant duplications |
| Lumpy complex | 387,898 | 60,130 | Number of complex (multiple tandem deletions or duplications) structural variants |
| BAC ends to same scaffold | 7,357 | 15,795 | BAC end pairs that were best mapped to the same scaffold |
| BAC ends to different scaffold | 21,228 | 12,791 | BAC end pairs with best alignments to different scaffolds |
| BAC ends unmapped | 484 | 483 | Unmapped BAC end pairs |

TGACv2 reference to the ARS_RCv1.1 assembly was performed with minimap2 [23] using the "-x asm10" preset. A Circos plot [43] derived from these alignments revealed numerous differences in sequence attribution to linkage group super-scaffolds (Figure 5A). Furthermore, these whole-scaffold alignments revealed several structural variants representing potential expansions of the TGACv2 reference compared to ARS_RCv1.1 (Figure 5B). The accuracy of ARS_RCv1.1 super-scaffold placement on a macro-scale was examined by alignment of previously generated BAC end sequence data from the Milvus B individual [14] to both assemblies with minimap2 using the "-x sr" preset. Resulting PAF files were analyzed with custom scripts [44] to identify three distinct categories of BAC paired-end alignments: (1) if both pairs aligned to the same scaffold, (2) if both pairs aligned to different scaffolds or (3) if both pairs were unmapped (Table 3). The same 483 BAC paired ends were unmapped to both assemblies, suggesting contamination in the original BAC library. However, the ARS_RCv1.1 assembly had two-fold more BAC paired-ends that aligned to the same super-scaffold than the TGACv2 reference. This gives greater confidence to the linkage-group assignment on the ARS_RCv1.1 assembly, and suggests that observed structural expansions of the TGACv2 reference are caused by misassemblies (Table 2) or other smaller errors (Figure 4).

## Re-use potential

We report a new red clover reference assembly using a combination of HiFi and Nanopore-based long-read sequencing, with Omni-C and BAC-end sequence scaffolding to produce chromosome-scale super-scaffolds. The quality of the assembly demonstrates that low-error rate long reads are suitable for resolving issues in assembling allogamous heterozygous (>50%) diploid plant genomes and generating continuous scaffolds. The addition of Omni-C read linkage data supported generation of an assembly with only 143 scaffolds. These scaffolds were then combined into seven final linkage-group super-scaffolds, better reflecting the haploid structure of red clover chromosomes. Compared with a previous reference for the species, ARS_RCv1.1 contains 20% more assembled sequence and has an error rate that is lower by three orders of magnitude.

Comparative mapping statistics to other legume genome assemblies suggest that this assembly will enable better alignment of red clover short-read WGS data, improve gene model prediction, and facilitate transcriptomic studies and gene discovery efforts based on both marker–phenotype association and sequence identity. Previous assemblies of red clover were limited by the error-rates or length of reads used to construct them.

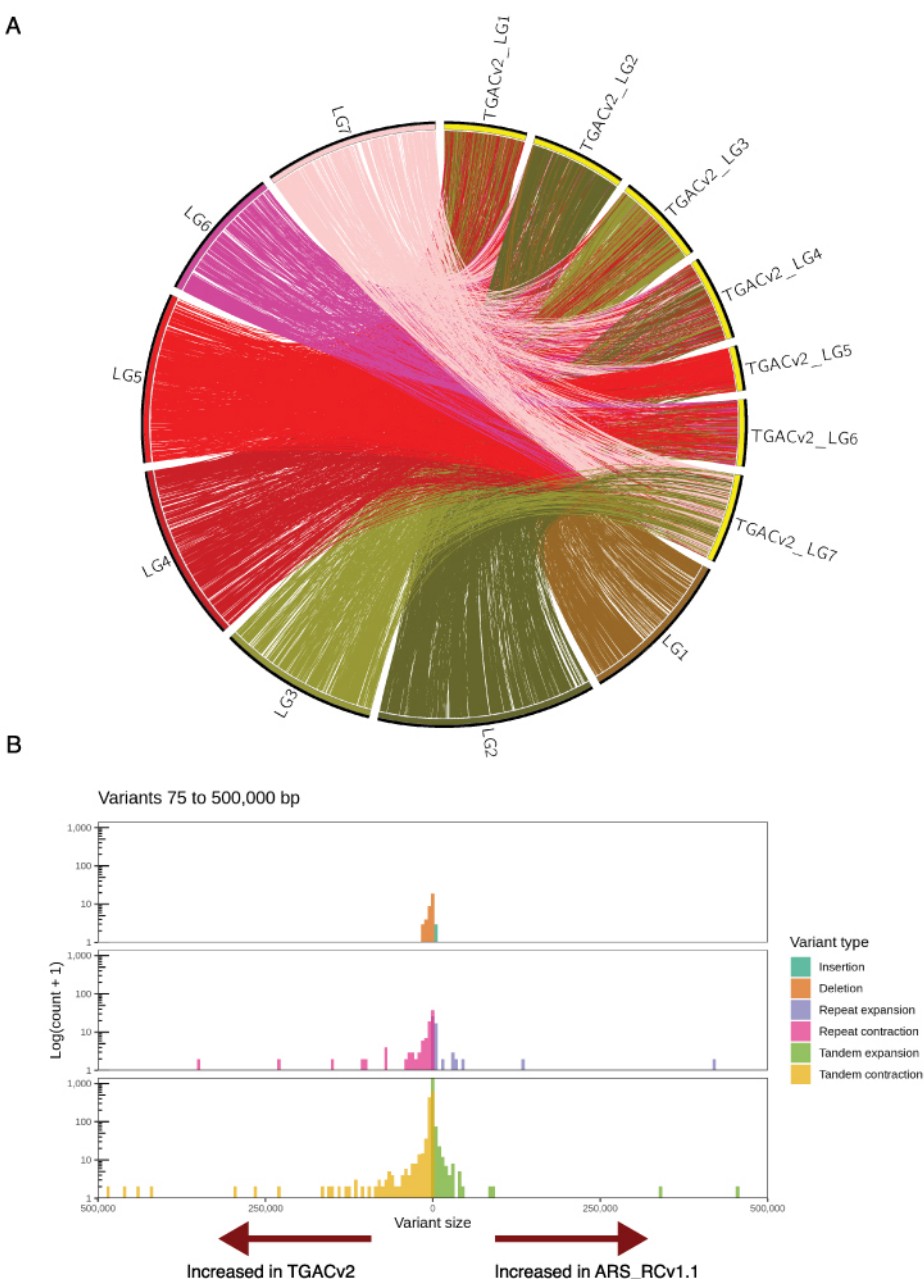

**Figure 5.** Structural variation comparison between the TGACv2 and ARS_RCv1.1 reference assemblies. (A) A Circos plot constructed from whole-genome alignments of TGACv2 (labelled TGACv2_LG1-7) to ARS_RCv1.1 (labelled LG1-7) is color-coded based on originating ARS_RCv1.1 linkage-group information. Only alignment blocks larger than 10 Kbp in length are displayed on the plot as ribbons that connect between each assembly. Presence of more than one colored alignment ribbon link to the TGACv2 scaffolds indicates a discrepancy between the two assemblies. (B) Whole-genome alignments also revealed additional structural variant discrepancies between the two assemblies. Given the relative nature of duplications and deletions detected on comparative alignments, arrows showing potential expansion of sequence in one assembly compared to another are indicated at the bottom of the plot. For example, tandem contractions of sequence in ARS_RCv1.1 could be considered expansions of genome sequence in TGACv2, and vice versa.

We demonstrate that recent improvements in DNA sequencing technologies are finally capable of generating a suitable assembly for this highly heterozygous species, and that these methods can be applied to other similar species without the need for expert curation.

## AVAILABILITY OF SOURCE CODE AND REQUIREMENTS

Project name: Themis-ASM.

Project Home page: https://github.com/njdbickhart/Themis-ASM.

Operating systems: Unix, Linux.

Programming language: Snakemake v3.4+, Python 3.6+, Perl 5.10+

Other requirements: miniconda v3.6+ or Anaconda 3+

License: GNU GPL

## DATA AVAILABILITY

All sequence data used in the assembly, scaffolding, and analysis of ARS_RCv1.1 can be found in the Sequence Read Archive (SRA) of the National Center for Biotechnology Information (NCBI) under Bioproject accession number PRJNA754186. Genome Accession for the ARS_RCv1.1 assembly is GCA_020283565.1. Iso-seq reads are in the NCBI SRA with run accession number SRR15433788. Iso-seq transcripts and other data are available via the Gigascience database, GigaDB [44].

## DECLARATIONS
## LIST OF ABBREVIATIONS

BAC: bacterial artificial chromosome; EST: expressed sequence tag; FLNC: full-length non-concatemer; Gbp: gigabase pairs; GSI: gametophytic self-incompatibility; Kbp: kilobase pairs; Mbp: megabase pairs; QV: quality value; SV: structural variant

## ETHICAL APPROVAL

Not applicable.

## CONSENT FOR PUBLICATION

Not applicable.

## COMPETING INTERESTS

The authors declare that they have no competing interests.

## FUNDING

This work was supported by USDA-ARS Projects 5090-31000-026-00D (DMB), 5090-21000-071-00D (MLS), 5090-21000-001-00D (HR), 3040-31000-100-00D (TPLS).

## AUTHORS' CONTRIBUTIONS

LMK, TPLS, and MLS generated the genome WGS, Omni-C, and transcriptome sequencing data. DMB and TPLS assembled the genome and DMB ran scaffolding analysis. DMB and LMK ran the assembly analysis. All authors read and approved the final version of the manuscript.

## ACKNOWLEDGEMENTS

We thank Dr Kristen Kuhn, Kelsey McClure, and Dr Jennifer McClure for technical assistance.

The USDA does not endorse any products or services. Mentioning of trade names is for information purposes only. The USDA is an equal opportunity employer.

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
